# Interplay between alkali-metal cations and silanol sites in nanosized CHA zeolite and implications for $CO_2$ adsorption

Sajjad Ghojavand [1], Eddy Dib[1], Jérôme Rey[2], Ayoub Daouli[2], Edwin B. Clatworthy [1], Philippe Bazin[1], Valérie Ruaux[1], Michael Badawi [2] & Svetlana Mintova [1✉]

Silanols are key players in the application performance of zeolites, yet, their localization and hydrogen bonding strength need more studies. The effects of post-synthetic ion exchange on nanosized chabazite (CHA), focusing on the formation of silanols, were studied. The significant alteration of the silanols of the chabazite nanozeolite upon ion exchange and their effect on the $CO_2$ adsorption capacity was revealed by solid-state nuclear magnetic resonance (NMR), Fourier-transform infrared (FTIR) spectroscopy, and periodic density functional theory (DFT) calculations. Both theoretical and experimental results revealed changing the ratio of extra-framework cations in CHA zeolites changes the population of silanols; decreasing the $Cs^+/K^+$ ratio creates more silanols. Upon adsorption of $CO_2$, the distribution and strength of the silanols also changed with increased hydrogen bonding, thus revealing an interaction of silanols with $CO_2$ molecules. To the best of our knowledge, this is the first evidence of the interplay between alkali-metal cations and silanols in nanosized CHA.

[1] Normandie Université, ENSICAEN, UNICAEN, CNRS, Laboratoire Catalyse et Spectrochimie (LCS), 14000 Caen, France. [2] Université de Lorraine, CNRS, Laboratoire de Physique et Chimie Théoriques (LPCT), F-54000 Nancy, France. ✉email: svetlana.mintova@ensicaen.fr

Excessive $CO_2$ admission to the atmosphere is one of the main contributors to current anthropogenic global warming[1,2]. $CO_2$ capture and separation technologies namely include (1) absorption with amine-based solvents, (2) adsorption by nanoporous solids, (3) cryogenic distillation, and (4) membranes[3–6]. Among all these technologies, $CO_2$ adsorption by zeolites has the advantages of being inorganic non-toxic materials with high stability due to their crystalline structure and high $CO_2$ selectivity tunable through their chemical composition and structural features[7–11]. Chabazite (CHA) zeolite with low Si/Al ratios is among the most promising solid adsorbents for the separation of $CO_2$ from mixtures containing $CH_4$ or $N_2$[9,10,12]. High-Al-containing CHA zeolites demonstrated high adsorption selectivity toward $CO_2$ over $N_2$ or $CH_4$ due to the ability of extra-framework cations (EFCs) to regulate guest molecule admission[9,12–14]. Initially described as molecular trapdoor behavior by Shang et al. for micron-sized CHA zeolites, this behavior occurs due to the occupation of the zeolite eight-membered rings (8MR) by EFCs which govern access to the internal zeolite pore network. The EFCs located within the 8MRs selectively reject or admit different guest molecules based on the respective cation-molecule interactions and thermal oscillation of the EFCs at a given temperature[12–14].

In theory, conventional CHA zeolites suffer from diffusion limitations of guest molecules through their pore networks due to being composed of polycrystalline particles with an average size of several micrometers[15,16]. Several methods have been developed to overcome these limitations by increasing the surface/volume ratio[15–17]. Among the known microporous materials, nanosized CHA zeolites are attractive candidates as they are composed of discrete nanoparticles (single crystals), giving rise to a higher external surface area and greater availability of active sites compared to the micron-sized CHA[15,16,18,19]. Recently, we reported on the $CO_2$ single-gas adsorption behavior of nanosized CHA samples with a particle size of 60 nm and a Si/Al ratio of 2.2 with different alkali-metal cations by combining classical modeling, experimental adsorption measurements, and in situ Fourier-transform infrared (FTIR) spectroscopy[9]. In this study, the nanosized CHA zeolites containing $K^+$ and $Cs^+$ cations have demonstrated trapdoor behavior; however, the large $Cs^+$ cations significantly restricted the available spaces within the CHA cages for $CO_2$ guest molecules[9].

Although the two proposed mechanisms, i.e., molecular sieving and trapdoor behavior, have significantly contributed to the understanding and optimization of $CO_2$ performance in zeolites, a strict relationship is yet to be found. This may be due to a missing piece of information on the nature of potential adsorption sites, in particular zeolites. Specifically, it is already known that the adsorption of $CO_2$ on various amorphous silica nanoparticles is partially due to the hydrogen bonding interaction between silanol sites and $CO_2$ guest molecules resulting in significant $CO_2$ adsorption capacities[20–23]. For instance, Venet et al. studied the $CO_2$ adsorption behavior of hydrophilic and partially hydrophobized (methylated) microporous silica[21]. The $CO_2$ capacity of the hydrophilic sample enriched in silanol sites was ~3.0 mmol g$^{-1}$ at 1000 kPa $CO_2$ and 303 K, which is almost two times higher compared to the partially hydrophobized sample with fewer available silanol sites (~1.6 mmol g$^{-1}$ at 1000 kPa $CO_2$ and 303 K)[21]. These results invite questions about how silanol sites may participate in $CO_2$ adsorption on zeolitic materials. FTIR analysis has been employed frequently to interrogate the interaction of $CO_2$ with zeolites; however, the presence of different adsorption sites (EFCs, Brønsted acid sites), variations of the carbonate adsorption modes, and the effects of temperature and trace amounts of $H_2O$ make clear interpretation of the analysis non-trivial[24]. In particular, the overlap of the $CO_2$ combination modes in the O–H region further complicates any clear determination of $CO_2$-silanol interactions. In earlier FTIR studies, surface OH groups have been invoked in the formation of carbonates on zeolites, such as 13X[25], as well as being implicated in weak interactions with $CO_2$ on Na-Y zeolite[26]. Furthermore, Delgado and Areán observed a perturbation of the silanol band attributed to the interaction of free silanols with $CO_2$ on H-Beta[27]. Recently, Rzepka et al. suggested that the silanol sites may participate in $CO_2$ adsorption on zeolite Na-A by forming chemisorbed species[28]. $^1H$ MAS NMR spectra of Na-A zeolite equilibrated with 1 bar of $^{13}CO_2$ revealed significant changes in the O–H region. This was rationalized with the formation of $HCO_3^-$ at the expense of either residual $H_2O$ and/or OH groups. Changes in the O–H stretching region in the presence of $CO_2$ observed by in situ FTIR spectroscopy were also taken to support this conclusion[28].

In this work, we focus on understanding the nature of silanol sites in different alkali-metal forms of CHA zeolite nanocrystals ($K^+$, $Cs^+$, and mixed) in the absence and presence of $CO_2$ using high-resolution solid-state magic-angle spinning nuclear magnetic resonance (MAS NMR) combined with in situ FTIR spectroscopy. The Al and Si environments of nanosized CHA samples were initially characterized by $^{27}Al$ and $^{29}Si$ MAS NMR spectroscopy; a constant Si/Al ratio and similar Si and Al environments were observed for all samples. In situ FTIR and $^1H$ MAS NMR analysis revealed clear changes to the total number of silanols and the distribution of their hydrogen bonding strength in response to changes in the EFC content of the nanosized CHA zeolites. A decrease in the $Cs^+/K^+$ ratio increased the total amount of silanol sites, while periodic DFT calculations showed significantly different formation energies of silanols in the 8MR occupied by either $Cs^+$ or $K^+$. These results suggest increasing the $Cs^+$ EFC content promotes silanol site healing. FTIR and $^1H$ MAS NMR spectra before and after $CO_2$ adsorption revealed a redistribution of the hydrogen bonding energies; the distribution of moderately hydrogen bonding sites increases at the expense of weak hydrogen-bonded sites in the presence of adsorbed $CO_2$.

## Results and discussion

Three different alkali-metal forms of nanosized CHA were studied: (a) as-prepared nanosized CHA zeolite, abbreviated as AP-CHA, (b) K-CHA, and (c) Cs-CHA (see Table 1 for exact chemical compositions). The chemical compositions of the samples are summarized in Table 1. The CHA nanocrystals containing

**Table 1 Chemical formula and silanol concentration of activated nanosized AP-CHA, K-CHA, and Cs-CHA zeolite samples determined by in situ FTIR and $^1H$ MAS NMR spectroscopy.**

| Sample | Chemical formula | FTIR $C_{OH}$/cm$^{-1}$ g$^{-1}$ | $^1H$ MAS NMR $C_{OH}$/mmol g$^{-1}$ | $\varepsilon_{OH}$* cm mmol$^{-1}$ |
|---|---|---|---|---|
| AP-CHA | $Na_{1.8}K_{5.7}Cs_{4.0}Al_{11.1}Si_{24.8}O_{72}$ | 644 | 3.7 | 348 |
| K-CHA | $K_{10.4}Cs_{1.0}Al_{11.4}Si_{24.9}O_{72}$ | 726 | 5.7 | 255 |
| Cs-CHA | $K_{1.8}Cs_{10.5}Al_{11.7}Si_{24.2}O_{72}$ | 531 | 3.4 | 316 |

*The molar absorption coefficients ($\varepsilon_{OH}$) are calculated using $\varepsilon_{OH} = A \, S / n_{OH}$ where A is the OH concentration based on FTIR, S is the area of pellets (2.01 cm$^2$), and $n_{OH}$ is the OH concentration based on $^1H$ MAS NMR.

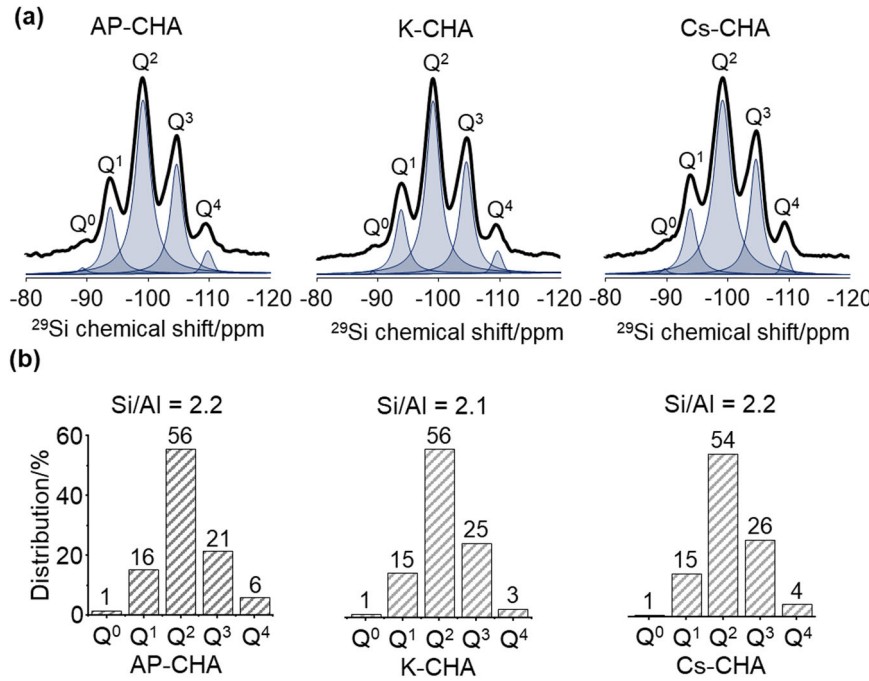

**Fig. 1 $^{29}$Si MAS NMR results of AP-CHA, K-CHA, and Cs-CHA nanosized zeolite samples at 298 K. a** The spectra and **b** the percentage of different $Q^n$ species.

different alkali-metal cations were characterized by powder X-ray diffraction (XRD), and the results are presented in Supplementary Fig. S3. The three patterns correspond to the CHA framework type zeolite[29].

$^{29}$Si MAS NMR spectra and the corresponding percentages of different $Q^n$ species ($n = Si[SiO]_n[X]_{4-n}$ where X = AlO or OH) are presented in Fig. 1a, b. All samples display five sharp resonances at −89.2, −93.9, −99.2, −104.7, and −109.4 ppm, consistent with the five distinct crystallographic $SiO_4$ sites assigned to $Q^{0-4}$ species, respectively[10,19]. These results are in accordance with the ones observed by Shang et al. for micron-sized CHA zeolites and our previous report for nanosized CHA zeolites[13,19]. The Si/Al ratio of the samples obtained by the deconvolution of the $^{29}$Si MAS NMR spectra (Fig. 1b) is the same after the ion exchange of the AP-CHA sample with $K^+$ and $Cs^+$ cations (2.1–2.2), consistent with the inductively coupled plasma mass spectrometry (ICP-MS) results presented in Table 1. $^{27}$Al MAS NMR spectra of the different nanosized alkali-metal forms of CHA contain a band at 57.7 ppm corresponding to tetrahedrally coordinated Al (Supplementary Fig. S4); no band at 0 ppm associated with the presence extra-framework Al species was detected[13,19]. The Al distribution within the different alkali forms of nanosized CHA samples remains the same upon ion exchange (Fig. 1a, b and Supplementary Fig. S4).

Figure 2 presents the silanol region of the FTIR spectra of the activated nanosized CHA samples under the adsorption of $CO_2$ at 27.2 kPa at equilibrium. The activated nanosized CHA contains a wide range of silanol sites (3100–3775 cm$^{-1}$, Fig. 2a) tentatively separated by different colors based on the strength of the hydrogen bonding of these sites in the FTIR spectra (vide infra). Exact identification and quantification of the different silanol sites by FTIR spectroscopy are not possible due to the complex nature of the O–H stretching frequency in the silanol region[30–33]. However, the overall concentration of silanol sites in the activated samples was estimated by integration of the band areas in the region of 3100–3775 cm$^{-1}$; the samples show different concentrations of silanol sites following the tendency K-CHA > AP-CHA > Cs-CHA (Table 1). FTIR spectra of the nanosized CHA

zeolite samples loaded with 27.2 kPa of $CO_2$ in equilibrium are shown in Fig. 2b. All the bands corresponding to the silanol sites shift by approximately 7 cm$^{-1}$ after adsorption of 27.2 kPa $CO_2$ at equilibrium (Fig. 2b). As an example, the band assigned to the isolated silanol sites (3738–3742 cm$^{-1}$) shifts toward lower wavenumbers (3733–3738 cm$^{-1}$), suggesting certain hydrogen bonding at these sites (Fig. 2a vs. Fig. 2b)[30,33]. Capturing the changes in the silanol region after the $CO_2$ adsorption is hindered by the $CO_2$ bands at 3700 and 3595 cm$^{-1}$ assigned to $v_3 + v_1$ and $v_3 + 2v_2$ modes of adsorbed $CO_2$ molecules, respectively ($v_1$ is the symmetric stretching mode and $v_3$ is the asymmetric stretching mode of $CO_2$)[34].

To be able to study the silanol groups more closely, high-resolution $^1$H MAS NMR spectra were acquired for the activated zeolite samples prior to and after $CO_2$ loading. The proton spectra and the corresponding deconvolutions are shown in Fig. 3 and Supplementary Fig. S5. The total concentration of the silanol sites of the activated nanosized CHA zeolite samples based on $^1$H MAS NMR spectroscopic analysis is summarized in Table 1. No signals related to the formation of bicarbonate species were observed upon adsorption of $CO_2$ guest molecules[28]. Four distinct regions of silanol sites were identified (Fig. 3 and Supplementary Fig. S5): (1) silanol sites interacting with cations (in purple); FTIR O–H stretching frequency in the range of 3766–3756 cm$^{-1}$ and negative proton chemical shifts, (2) isolated or weakly hydrogen-bonded silanol sites (in blue); FTIR O–H stretching frequency in the range of 3756–3700 cm$^{-1}$ and proton chemical shift of 0.0–2.0 ppm, (3) moderately hydrogen-bonded silanol sites (in yellow); FTIR O–H stretching frequency in the range of 3700–3470 cm$^{-1}$ and proton chemical shift of 2.0–4.5 ppm, and (4) strongly hydrogen-bonded silanol sites (in red); FTIR O–H stretching frequency in the range of 3470–3100 cm$^{-1}$ and proton chemical shifts above 4.5 ppm[33]. Generally, as the proton chemical shift increases, the O–H stretching frequency decreases, and the strength of the hydrogen bonding increases[30,33]. The total concentration of silanol sites changes upon ion exchange with $K^+$ and $Cs^+$ alkali-metal cations. The AP-CHA sample presents an overall silanol concentration of

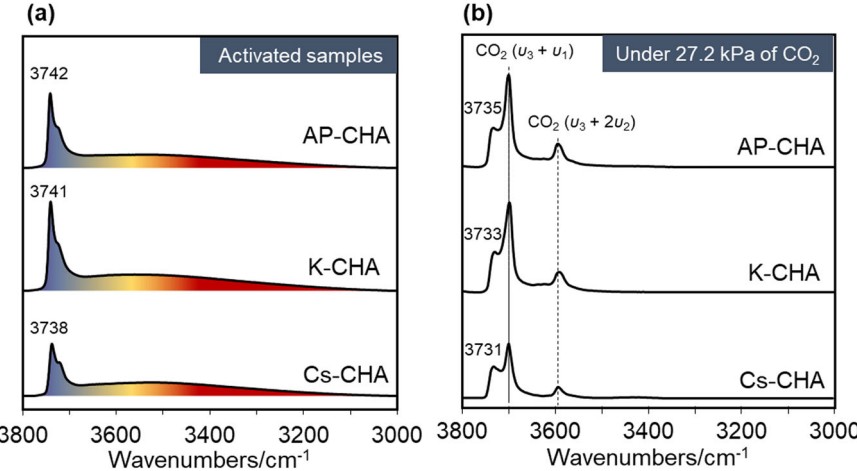

**Fig. 2 FTIR spectra of silanol region (3000–3800 cm$^{-1}$) of nanosized AP-CHA, K-CHA, and Cs-CHA zeolite samples at 298 K. a** Samples pretreated at 623 K under high vacuum ($10^{-6}$ kPa) prior to measurement and **b** at 27.2 kPa $CO_2$ equilibrium. Color code: purple for silanol sites interacting with cations, blue for isolated and weakly hydrogen-bonded, yellow for moderately hydrogen-bonded, and red for strongly hydrogen-bonded silanol sites. Peak assignment: isolated silanol sites ~3740 cm$^{-1}$, $\upsilon_3 + \upsilon_1$ of $CO_2$ at 3700 cm$^{-1}$, and $\upsilon_3 + 2\upsilon_2$ of $CO_2$ at 3595 cm$^{-1}$.

3.7 mmol g$^{-1}$ based on $^1$H MAS NMR (Table 1). Upon ion exchange with Cs$^+$ (Cs-CHA sample), the silanol concentration drops to 3.4 mmol g$^{-1}$, while for the K-CHA sample, the total silanol concentration increases to 5.7 mmol g$^{-1}$ based on $^1$H MAS NMR study (Table 1). Both $^1$H MAS NMR and FTIR results show similar trends regarding the overall concentration of silanol sites (Table 1).

The total concentration of the silanol sites in the different nanosized CHA samples changes with the content of the K$^+$ or Cs$^+$ cations across high, intermediate, and low values (Fig. 4). These results clearly demonstrate that performing ion exchange even at mild conditions (at neutral pH and room temperature) can significantly alter the concentration of the silanol sites.

The significant changes in the total silanol concentrations for the Cs-CHA and K-CHA samples of 3.4 and 5.7 mmol g$^{-1}$, respectively, are shown (Table 1). Following the spectroscopic results, periodic DFT calculations were carried out to elucidate the energy of silanol formation of Cs-CHA and K-CHA. Three different cases were chosen to study the energy of silanol formation in K-CHA and Cs-CHA samples: (1) silanol in a 6MR, (2) silanol in the confined corner of an 8MR next to a 4MR, and (3) silanol in an 8MR (see Fig. 5).

Silanol formation was investigated initially in confined sites of the K-CHA and Cs-CHA samples. In the 6MR, similar behavior for the K-CHA and Cs-CHA samples was observed with estimated electronic reaction energies of 3 and −3 kJ mol$^{-1}$, respectively (Fig. 5a). Analogously, similar reaction energies of 29 and 21 kJ mol$^{-1}$ were obtained for K-CHA and Cs-CHA, respectively, in the confined corner of the 8MR next to 4MR site (Fig. 5b). In comparison, only the most accessible part of the 8MR site (Fig. 5c) showed significantly different energies for silanol formation of 51 and 83 kJ mol$^{-1}$ for K-CHA and Cs-CHA, respectively. Overall, the results presented in Fig. 5 show that the confined sites of the chabazite zeolite structure exhibit similar silanol formation energies for both K-CHA and Cs-CHA samples. In contrast, at the most accessible site in the 8MR, as demonstrated in our previous work for Cs$^+$ cations[10], the formation of silanol defects is demonstrated to be more feasible in the presence of K$^+$ than Cs$^+$ cations (Fig. 5c). Thus, the DFT results support the experimental findings; this can be attributed to a stronger interaction of the smaller and more electronegative K$^+$ cation with O atoms in the zeolite framework[35]. We have already observed that large Cs$^+$ cations with ionic radii of 1.67 Å compared to K$^+$ with ionic radii of 1.37 Å are promoting the

formation of long-range crystalline order and stabilization of the CHA structure[19,36]. The theoretical and experimental findings in the current work show that Cs$^+$ cations introduced by ion exchange also stabilize the CHA framework (Table 1, Figs. 4 and 5). Thus, the generation of 68% additional structural defects can be explained by the decrease of the Cs$^+$ EFC content in the unit cell by 90% (Fig. 4 and Table 1).

$CO_2$ in similar positions has been considered for K-CHA and Cs-CHA zeolite samples. The three configurations are reported in Supplementary Table S1. The most favorable structure (case c) is shown in Supplementary Fig. S6. In all cases, the adsorption of $CO_2$ is more favorable for sample K-CHA (48–59 kJ mol$^{-1}$ in absolute value) than for the Cs-CHA (Supplementary Table S1). Especially in the most favorable configuration (case c), $CO_2$ adsorption energies of −57 and −54 kJ mol$^{-1}$ are obtained for K-CHA and Cs-CHA, respectively. This trend is consistent with our previous findings for the isosteric heat of $CO_2$ adsorption that are −41 and −35 kJ mol$^{-1}$ for K-CHA and Cs-CHA, respectively[9].

$^1$H NMR chemical shifts were also calculated, and the results are summarized in Supplementary Table S2. Upon adsorption of $CO_2$, the chemical shifts of silanols' are changing in most cases, reflecting the evolution of the hydrogen bonding strength within both structures in the presence of $CO_2$. As shown, the theoretical results are in line with the experimental observations. Moreover, upon exposing the chabazite samples to 27.2 kPa $CO_2$ at equilibrium, the peaks corresponding to isolated and weakly hydrogen-bonded silanols in the $^1$H MAS NMR spectrum (in blue) slightly shift by 0.1 ppm to higher values (Fig. 3 and Supplementary Fig. S5). This tendency is also observed by FTIR, i.e., the band corresponding to isolated silanol sites at ~3740 cm$^{-1}$ shifts by 7 cm$^{-1}$ to lower wavenumber; this result is in good accordance with the correlations shown in our previous work (Fig. 2 vs. Supplementary Fig. S5)[33]. All samples clearly display a 0.1 ppm shift of the blue peaks in the $^1$H MAS NMR spectra at 27.2 kPa $CO_2$: a shift from 1.2 to 1.3 ppm for sample AP-CHA, from 0.9 to 1.0 for sample K-CHA, and from 1.1 to 1.2 for sample Cs-CHA was measured (Supplementary Fig. S5). The results suggest an increase in the hydrogen bonding strength after $CO_2$ adsorption. The $^1$H MAS spectra of the samples redistribute toward higher chemical shifts, as shown in Supplementary Fig. S5. Based on our previous IR study, we found up to 3.5% chemisorbed $CO_2$ on different alkali-metal forms of nanosized CHA[9]. The

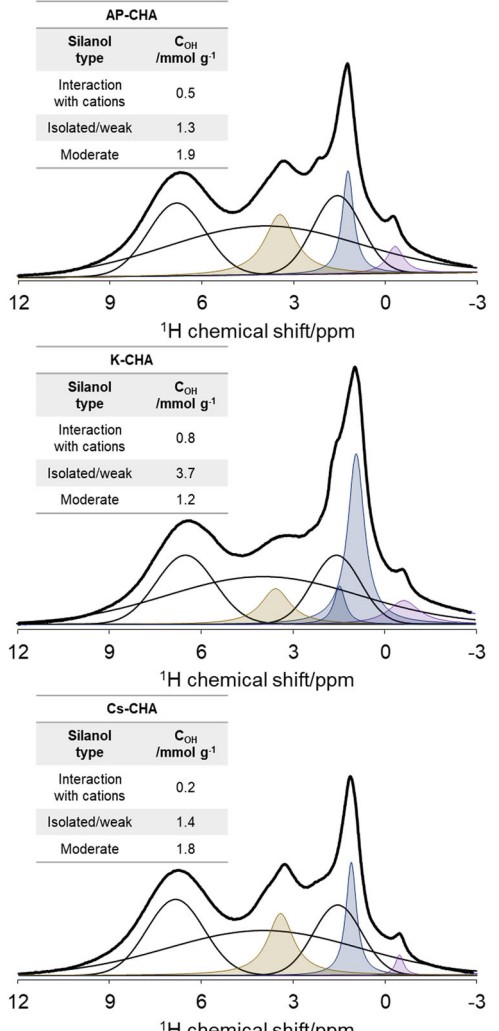

**Fig. 3 $^1$H MAS NMR spectra of nanosized AP-CHA, K-CHA, and Cs-CHA zeolite samples.** All samples were pretreated at 623 K under high vacuum ($10^{-6}$ kPa) prior to measurements. Color code: rotor signals in hollow black lines, purple for silanol sites interacting with cations, blue for isolated and weakly hydrogen-bonded, and yellow for moderately hydrogen-bonded silanol sites. The intensity of the rotor signal (slightly shifting due to differences in magnetic susceptibilities of the samples) is kept constant during the fitting procedure for all samples. Then, small errors are expected in the quantification of the silanol sites.

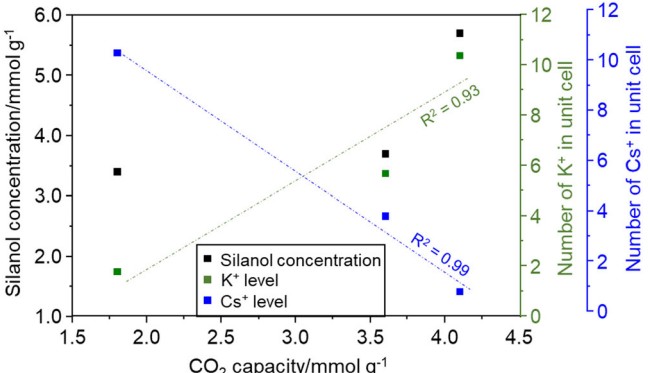

**Fig. 4 $CO_2$ capacity vs. the concentration of silanols, $K^+$, and $Cs^+$.** Dashed lines show the linear goodness of fits.

chemisorbed species were assigned to the formation of free, monodentate, bidentate, and bridged bidentate carbonate species[9]. However, unlike the work of Rzepka et al. on studying $CO_2$ adsorption on Na-A zeolite, we do not see any resonances at 12 and 16 ppm in $^1$H MAS NMR spectra assigned to bicarbonate species[28]. Higher $CO_2$ capacities were recorded when silanol site concentration increased (Fig. 4). While it should be noted that $Cs^+/K^+$ cationic ratios are also changing, the cations are indeed the main adsorption sites (Fig. 4)[10]. However, the changes observed in the $^1$H MAS NMR spectra toward higher H-bond strengths upon $CO_2$ adsorption (Supplementary Fig. S5) suggest that silanol sites are sensitive to the adsorption of $CO_2$ on the different alkali-metal forms of nanosized CHA samples. The exact magnitude of this contribution is not yet evident, the quantification of silanol sites being not obvious when dipolar couplings occur despite the use of high MAS rates[37]. In summary, increasing the content of $Cs^+$ EFCs clearly shows a healing effect on the silanol sites (up to 68%). To the best of our knowledge, this is the first work revealing the interplay between alkali-metal cations and silanol sites in nanosized zeolites made possible by high-resolution solid-state $^1$H MAS NMR combined with FTIR spectroscopy.

## Conclusions

In conclusion, we have investigated the silanol sites of different alkali-metal forms of nanosized CHA zeolites by combining in situ FTIR and $^1$H MAS NMR spectroscopy in the absence and presence of $CO_2$ guest molecules. Using $^1$H MAS NMR spectroscopy, the quantification and assignment of four distinct regions of silanol sites based on the strength of hydrogen bonding was performed. Ion exchanging nanosized CHA zeolites with $K^+$ and $Cs^+$ significantly altered the silanol distribution revealing a new avenue for tailoring zeolite properties. This is due to the stabilizing role of $Cs^+$ cations in the CHA structure; by decreasing the $Cs^+$ content from 10 to 1 cation per unit cell (increasing the $K^+$ content), an increase of the total silanol concentration from 3.4 to 5.7 mmol g$^{-1}$ was found; the experimental results were further supported by DFT calculations showing $Cs^+$ cations in the 8MR significantly decreased the energy of silanol formation. The complex nature of the O–H stretching frequency in the region of 3000–4000 cm$^{-1}$ of FTIR spectra and the overlapping with complex $CO_2$ stretching modes hinders the study of silanol sites using this method. However, the $^1$H MAS NMR spectroscopy can shed light on the nature of the silanol sites in zeolites. Upon adsorption of $CO_2$ (27.2 kPa) on different alkali-metal forms of nanosized CHA zeolite, the distribution of the strength of silanol sites within the CHA framework was altered. The strength of hydrogen bonding increased, thus demonstrating the interaction of silanols with $CO_2$ molecules. From these results, the development of new methods to engineer specific silanol sites could be a prospective area for improving the $CO_2$ adsorption performance of zeolites.

## Methods

**Materials and synthesis**. Nanosized CHA samples were synthesized based on our previously published works[10,19]. All the water used during the synthesis and purification was double distilled water. First, 0.54 g of sodium aluminate (NaAlO$_2$, 40–45% Na$_2$O, 50–56% Al$_2$O$_3$, Sigma-Aldrich), 1.7 g of sodium hydroxide (NaOH, 98 wt%, Sigma-Aldrich), 0.824 g of potassium hydroxide (KOH, 85 wt%, Sigma-Aldrich), 0.442 g of cesium hydroxide (CsOH, 50 wt.% Cs in water, Alfa Aesar), and 4.2 g of water were mixed for 2 h to obtain a clear solution. Afterward, 10 g of colloidal silica (LUDOX AS-40, Sigma-Aldrich) was added dropwise under vigorous stirring to obtain a composition of the precursor mixture of 0.2 Cs$_2$O: 1.5 K$_2$O: 6.0 Na$_2$O: 16.0 SiO$_2$: 0.7 Al$_2$O$_3$: 141.7 H$_2$O. The mixture was aged for 17 days at room temperature under vigorous stirring, followed by hydrothermal treatment at 90 °C for 7 h. The crystalline CHA samples were recovered by multi-step centrifugation and washed with water to remove all unreacted reagents. The recovered CHA nanocrystals were dried in an oven at 60 °C overnight, and the final

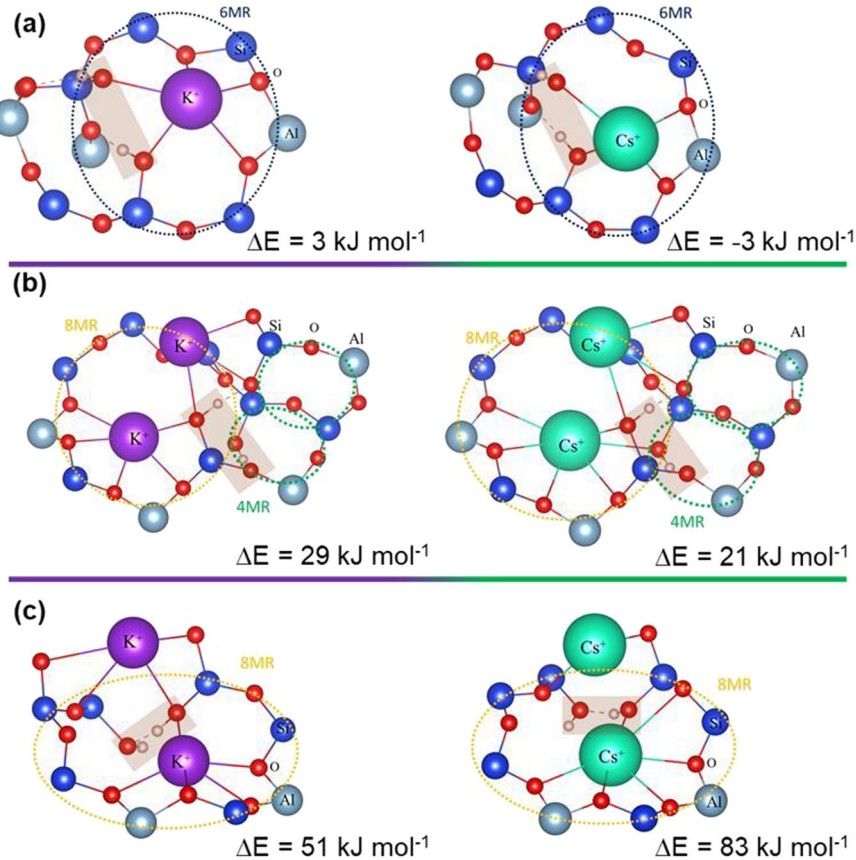

**Fig. 5 DFT cases studied for silanol formation of K- and Cs-CHA and their electronic energies. a** A 6MR, **b** the confined corner of an 8MR next to a 4MR, and **c** an 8MR. Silanol sites are highlighted by shallow brown boxes, 8MR is highlighted by orange, 6MR by dark blue, and 4MR by green circles. Color code: Si in blue, O in red, Al in gray, K in purple, and Cs in green.

as-prepared CHA nanocrystals were labeled as AP-CHA. AP-CHA sample was ion-exchanged with 1 M salts of potassium chloride or cesium chloride (KCl, 99 wt %, Sigma-Aldrich and CsCl, 99.5 wt%, Alfa Aesar) at a liquid/solid ratio of 40 mL g$^{-1}$ for 2 h at room temperature; this procedure was repeated three times. The final samples were recovered by centrifugation and dried at 60 °C overnight. The ion-exchanged samples with K$^+$ and Cs$^+$ were labeled as K-CHA and Cs-CHA, respectively.

**Characterization**. Powder X-ray diffraction (XRD) patterns were collected with a PANalytical X'Pert Pro diffractometer using Cu-Kα1 radiation (λ = 1.5406 Å, 45 kV, 40 mA). The patterns were collected between 2θ of 5 and 60° with a step size of 0.0167° and time per step of 1000 s. Inductively coupled plasma mass spectrometry (ICP-MS) measurements were performed using a 7900 ICP-MS from Agilent Technologies.

Magic-angle spinning nuclear magnetic resonance (MAS NMR) spectra of $^{29}$Si and $^{27}$Al nuclei were recorded with a single pulse on a Bruker Avance 500 MHz (11.7T) spectrometer using 4 mm-OD zirconia rotors with a spinning frequency of 12 and 14 kHz, respectively. Single pulse excitation (30° and 15° flip angle) of 3 µs was used for $^{29}$Si and $^{27}$Al MAS NMR experiment and 30 s of recycle delay. Tetramethylsilane (TMS) and aluminum nitrate Al(NO$_3$)$_3$ were used as references for $^{29}$Si and $^{27}$Al nuclei, respectively. One dimensional $^1$H MAS NMR spectra were acquired at 500.07 MHz on a Bruker Avance III-HD (11.7T), using 1.9 mm outer diameter probe zirconia rotors spun at 40 kHz, a radiofrequency power of 114 kHz and a recycle delay of 10 s. The modeling of $^1$H NMR spectra is done using Lorentzian peaks in Dmfit software[38]. The $^1$H MAS NMR spectra were collected both after activating the samples at 350 °C under vacuum overnight and after dosing the samples with 27.2 kPa of CO$_2$ at equilibrium using a Micromeritics 3Flex Surface Characterization unit (Norcross, GA, USA) which was back-filled with Ar to atmospheric pressure and the NMR rotor were closed under Ar in a glovebox. The proton concentration was calculated using adamantane as a reference. We used a 1.9 mm rotor in this work. For all $^1$H MAS NMR results presented in this work, the rotor signal was subtracted during the fitting of the spectra. The empty rotor signal is presented in the Supplementary Information (Supplementary Fig. S1). However, we should note that due to the change in the magnetic susceptibility of the rotor (empty vs. filled), slight chemical shift variations of the rotor signal occur when filled. As a result, the subtraction of the rotor signal is not a trivial task using the routine procedures available in Topspin

software. Hahn echo and anti-ringing experiments were not enough to fully suppress the rotor signal without losing a part of the broad signals corresponding to the samples (see Supplementary Figs. S7 and S8). Hence, despite the errors that may be induced by the fitting procedure, only single pulse experiments are discussed in this work, and the rotor signal was considered during the fitting procedure after small adjustments of chemical shifts while keeping the overall intensity of the rotor caps constant as it was done in our previous works[30,33,39,40]. Although this procedure is the current state of the art, the small adjustments can introduce different degrees of unknown errors during the silanol site quantifications. To the best of our knowledge, these errors are unavoidable when considering such amounts of protons that are involved in H bonds; a work is under investigation to improve the quantification of silanol sites.

Fourier-transform infrared (FTIR) spectroscopic measurements of nanosized zeolite samples were performed on a self-supported pellet (~20 mg and a diameter of 16 mm) on a Thermo Scientific Nicolet iS50 FTIR spectrometer with a spectral resolution of 4 cm$^{-1}$. For all the FTIR measurements, we used a homemade cell called Carroucell[41]. The specificity of this cell is to be able to measure 12 samples simultaneously in a single chamber at identical experimental conditions. Prior to each measurement, the samples were activated by heating to 350 °C with a ramp of 3 °C min$^{-1}$ under high vacuum (~10$^{-6}$ kPa), and the spectra of the activated samples were collected. All CO$_2$ FTIR spectra were recorded at room temperature under 27.2 kPa of CO$_2$ in equilibrium.

**Periodic density functional theory calculations**. Periodic density functional theory (DFT) calculations were done using VASP (Vienna Ab initio Simulation package)[42–44] with the semilocal Perdew–Burk–Ernzehof (PBE) functional and the projector-augmented wave (PAW) method developed by Blöchl[45,46]. Gaussian smearing of σ = 0.05 eV was applied. The plane wave cutoff was set to 400 eV for the relaxation of ionic positions and 800 eV for the initial cell relaxation. The Kohn–Sham equations were solved self-consistently with a convergence criterion of 10$^{-7}$ eV. The ionic positions were relaxed until all forces were smaller than 0.01 eV Å$^{-1}$. The sampling of the Brillouin zone was restricted to the Γ-point as we used a large supercell (vide infra).

Since long-range dispersion interactions are not included in semilocal DFT functionals, a correct description of these dispersion interactions is crucial for zeolite simulations. To this end, the Tkatchenko-Scheffler scheme with iterative

Hirshfeld partitioning (TS/HI) has been used[47–49]. Taking into account the ionic character of the system, TS/HI has been shown accurate to such systems[50,51].

**The structural model used in DFT calculations**. In order to compare the formation of silanol sites in the vicinity of $K^+$ or $Cs^+$, a large supercell of Chabazite (CHA) with two large cavities, one filled with $K^+$ cations and the other with $Cs^+$ cations, was prepared. The two model systems mimic the synthesized K-CHA and Cs-CHA samples while permitting a direct comparison of the final products (the supercell with a Si–O–Si bridge broken in two silanol sites).

A triclinic supercell of CHA with 24 T sites used in previous modeling works ($a = 12.712$ Å, $b = 13.711$ Å, $c = 9.365$ Å, $\alpha = 90.32°$, $\beta = 96.82°$, $\gamma = 89.93°$) has been doubled in the *c* direction to obtain a larger supercell with 48 T sites[52]. Sixteen silicon atoms were substituted by aluminum atoms, yielding a material with a Si/Al ratio of 2. Sixteen extra-framework cations ($8 K^+$ on one cavity and $8 Cs^+$ on the other) were added to ensure electroneutrality, giving the formula of the unit cell of $K_8Cs_8Al_{16}Si_{32}O_{96}$. This cell was fully relaxed at given lattice parameters: $a = 12.652$ Å, $b = 13.912$ Å, $c = 18.799$ Å, $\alpha = 90.01°$, $\beta = 99.26°$, $\gamma = 89.97°$. As described in previous works[9,10], the distribution of extra-framework cations was as follows: $K^+$ and $Cs^+$ occupy SII (close to the center of the six-membered ring (6MR) inside the cavity), SIII (next to a four-membered ring (4MR) of a double six-membered ring (*d6r*) inside the cavity) and SIII′ (at the center of an eight-membered ring (8MR)) sites. For clarity, the cell is presented in Supplementary Fig. S2.

The formation of silanol sites upon the addition of water was considered by producing a structural defect in the internal structure where the Si–O–Si has been broken and saturated by an $H^+$ on one side and $OH^-$ on the other side (Fig. 5). The chemical reaction is as follows:

$$\dots -Si-O-Si-\dots + H_2O \rightleftharpoons \dots -SiOH + HOSi-\dots \quad (1)$$

and the electronic energy of this reaction was computed with the following equation:

$$\Delta E = E_S - E_{H_2O} - E_{CHA} \quad (2)$$

where $E_S$ is the electronic energy of CHA zeolite with silanol sites, $E_{H_2O}$ is the energy of water in the gas phase, and $E_{CHA}$ is the energy of the initial supercell of CHA without any silanol sites.

The adsorption energies of $CO_2$ were also determined from static DFT calculations with the following equation:

$$\Delta E_{ads}(CO_2) = E_{S+CO_2} - E_S - E_{CO_2} \quad (3)$$

where $E_{CO_2}$ is the energy of $CO_2$ in the gas phase. Different initial positions of $CO_2$ were investigated for the systems investigated, giving similar final configurations and adsorption energies, the cavity of chabazite zeolite being sterically constrained by the presence of the cations, leaving little available space for $CO_2$.

Chemical shifts of silanol protons, with and without adsorbed $CO_2$, were computed by means of the linear response approach[53,54], as implemented in VASP, with a cutoff energy of 500 eV. The isotropic chemical shift $\delta_{iso}$ of the silanol protons was calculated with the following equation:

$$\delta_{iso} = \sigma_{ref} - \sigma_{iso} \quad (4)$$

where $\sigma_{iso}$ is the isotropic chemical shielding of the proton and $\sigma_{ref}$ is the average of the isotropic chemical shielding of the protons of tetramethylsilane (TMS) in vacuum.

## Data availability

The datasets generated during and/or analyzed during the current study are available from the corresponding author upon reasonable request.

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

## Acknowledgements

This study was co-funded by the European Union (ERC, ZEOLIghT, 101054004). Views and opinions expressed are, however, those of the author(s) only and do not necessarily reflect those of the European Union or the European Research Council. Neither the European Union nor the granting authority can be held responsible for them. Financial support from the Normandy Region through the Label of Excellence for the Centre of Zeolites and Nanoporous Materials (CLEAR) is acknowledged. This work was granted access to the HPC resources of TGCC under the allocation 2022-A0120810433 by the GENCI-EDARI project. J.R., A.D., and M.B. also acknowledge the financial support of the COMETE project (COnception in silico de Matériaux pour l'EnvironnemenT et l'Energie) cofounded by the European Union under the program "FEDER-FSE Lorraine et Massif des Vosges 2014–2022".

## Author contributions

S.G.: synthesis, analysis, validation, visualization, writing of original draft, review & editing; E.D.: analysis, validation, writing—review & editing; J.R.: theoretical study, writing—review & editing; A.D.: theoretical study, writing—review & editing; E.B.C.: validation, review & editing; P.B.: investigation, validation; V.R.: analysis; M.B.: Supervision of theoretical study, validation, writing—review & editing; S.M.: funding acquisition, project administration, resources, supervision, validation, writing—review & editing.

## Competing interests

The authors declare no competing interests.
