## [Peer Review File · Communications Chemistry]

Reviewers' comments:

Reviewer #1 (Remarks to the Author):

The manuscript “Interplay between alkali-metal cations and silanol sites in nano-sized CHA zeolite: implications for CO₂ adsorption” by Sajjad Ghojavand and co-workers, focuses on the formation of silanol groups (Si-O-H) in the post-synthetic ion-exchanged, nano-sized chabazite (CHA) zeolite samples. This work is interesting, and definitely could provide insights into rather unknown role of the silanol groups/defects in CO₂ adsorption phenomena, however, some major concerns have to be addressed before this work could be considered for publication.

1) My first point concerns the quality/validity of the collected ¹H MAS NMR data in order to support main observations and conclusions of the work. This is a major problem. For all three samples background corresponds to (roughly) 70% of the detected ¹H NMR signal intensity. What is even more worrying is that when inspecting closely the Fig 3. the background signals appear to change across the spectra. This does not allow for unambiguous quantification of proton species. Moreover, the background signals of empty rotors shown in Fig. S1 do not correspond to any of the claimed rotor signals (in grey) in Fig. 3. Neither the intensity ratios of the background signals at ~7 ppm and ~1.8 ppm match each other when comparing Fig. 3 and Fig. S1, nor the background signals at ~4 ppm are included in the fits of Fig. 3. Instead, these minor background signals at ~4 ppm are interpreted as “strong” silanol groups in Fig. 3!

Such pronounced ¹H background signals may result from the coil contamination with residues of the markings on the rotors for MAS reader and with dirt/fat from the rotors touched with fingers. I suggest that ¹H NMR spectra are recorded again after the probe stator assembly and the coil (and rotors as well) are cleaned thoroughly. Two data sets of proton spectra should be recorded to provide unambiguous assessment. The first (I) using single-pulse spectra followed by subtraction of the spectrum of an empty rotor acquired under otherwise identical conditions, and the second data set (II) acquired using a rotor-synchronized spin-echo sequence, which should remove probe background and provide flat baseline for easy quantification. Then by comparing results from I and II one can get robust assessments. For data set I the spectra after subtraction of the empty rotor signal should be shown (the rest in SI), so the reader can see how the fitted spectrum looks like and how it compares with experiment.

2) The second point concerns the interpretation of the data in Fig. 4, and hence the conclusions of this work. When reading Figure 4 it is clear that CO₂ uptake capacity depends primarily on the K⁺/Cs⁺ ratio. The smaller K⁺ ion interacts stronger with CO₂ molecules allowing stronger electrostatic interactions at closer distances, and being smaller and situated closer to the framework is in general less restrictive for diffusion through the material. So this is expected and generally known. On the other hand the

role of the silanol groups is less clear and this is the most interesting part of this work. Although there seem to be a linear trend, the final assessment on this has to wait until the robust ^1H MAS NMR data are available (see point 1).

3) In my opinion, DFT calculations included in this work are of little importance, since it is difficult to validate or relate these results to experimental data. On the other hand, it would be much more beneficial for this work to include ^1H NMR shifts calculations for different models to provide (in conjunction with experimental ^1H MAS NMR spectra) direct atomic-level insights into potential silanol groups environments in the material. I strongly encourage authors to do this; it would greatly elevate the impact of the paper when the experimental spectra could be directly related to distinct defects/structural motifs.

4) I suggests some general improvements in the manuscript to improve clarity:

abstract, lines 19-21, "content" should rather be changed to ratios of different cations

page 4, lines 112-114, the sentence is long and difficult to follow, I think it should be improved, it is important that information about samples considered in the manuscript is presented clearly

Table 1, it is somehow awkward that molar absorption coefficients (that relate to IR) are presented after the NMR data, whereas IR data are presented first; NMR should be presented first instead, IR data are not quantitative anyway

page 5, lines 127-131, "number of Si surrounding a SiO_4 site", it is not clear for a non-NMR reader, should be explained clearly, as well as a shift dependence in case of Al neighbours

Reviewer #2 (Remarks to the Author):

In this work, the authors investigated the influence of K^+ and Cs^+ on the distribution of silanol sites and the variations after the CO_2 adsorption by FTIR and NMR. They also revealed the relationship between total silanol concentration and CO_2 adsorption capacities. The authors have done a systematic work concerning silanol sites properties and CO_2 adsorption with interesting findings. While some points should be addressed before its publication.

1 Since silanol sites play a vital role in the CO_2 adsorption in this work, silanol sites are suggested to be added in the keyword

2 The first two paragraphs in introduction part are similar to the author's previous published work (ACS Appl. Nano Mater. 2022, 5, 4, 5578–5588).

3 In the DFT part, the authors calculated the silanol formation energy of three cases, (i) silanol in a 6MR, (ii) silanol in confined corner of an 8MR next to a 4MR, and (iii) silanol in an 8MR. Before drawing the conclusion "In contrast, at the most accessible site in the 8MR, the formation of silanol defects is demonstrated to be more feasible in the presence of K^+ than Cs^+ cations" (page 11), the authors should give some evidences or cite literatures to prove the 8MR is the most accessible site for CO_2 rather the other two cases.

4 How does the author obtain the conclusion "The current theoretical/experimental findings show that Cs^+ cations can also stabilize the CHA framework by ion-exchange insertion (Table 1, Fig. 4, and Fig. 5)" (page 11)?

5 It is indicated that "In small-pore zeolites with high cation contents, cage access is typically controlled by the size of the cation and the occupancy degree of the window." (Nature Chemistry, 2013, 5, 89-90). Considering the large radius of Cs^+ , does the author think the occupancy of the cage window by Cs^+ could be a reason for the decreasing trend of CO_2 adsorption capacity vs Cs^+ amount?

Dear reviewers,

The manuscript was revised taking into accounts the comments and the changes are highlighted in both the revised manuscript and supporting information attached.

Here is a point-by-point response to the reviewers' comments and concerns.

Reviewer #1:

"The manuscript "Interplay between alkali-metal cations and silanol sites in nano-sized CHA zeolite: implications for CO₂ adsorption" by Sajjad Ghajavand and co-workers, focuses on the formation of silanol groups (Si-O-H) in the post-synthetic ion-exchanged, nano-sized chabazite (CHA) zeolite samples. This work is interesting, and definitely could provide insights into rather unknown role of the silanol groups/defects in CO₂ adsorption phenomena, however, some major concerns have to be addressed before this work could be considered for publication."

Comment 1: My first point concerns the quality/validity of the collected ¹H MAS NMR data in order to support main observations and conclusions of the work. This is a major problem. For all three samples background corresponds to (roughly) 70% of the detected ¹H NMR signal intensity. What is even more worrying is that when inspecting closely the Fig 3. the background signals appear to change across the spectra. This does not allow for unambiguous quantification of proton species. Moreover, the background signals of empty rotors shown in Fig. S1 do not correspond to any of the claimed rotor signals (in grey) in Fig. 3. Neither the intensity ratios of the background signals at ~7 ppm and ~1.8 ppm match each other when comparing Fig. 3 and Fig. S1, nor the background signals at ~4 ppm are included in the fits of Fig. 3. Instead, these minor background signals at ~4 ppm are interpreted as "strong" silanol groups in Fig. 3!

Such pronounced ¹H background signals may result from the coil contamination with residues of the markings on the rotors for MAS reader and with dirt/fat from the rotors touched with fingers. I suggest that ¹H NMR spectra are recorded again after the probe stator assembly and the coil (and rotors as well) are cleaned thoroughly. Two data sets of proton spectra should be recorded to provide unambiguous assessment. The first (I) using single-pulse spectra followed by subtraction of the spectrum of an empty rotor acquired under otherwise identical conditions, and the second data set (II) acquired using a rotor-synchronized spin-echo sequence, which should remove probe background and provide flat baseline for easy quantification. Then by comparing results from I and II one can get robust assessments. For data set I the spectra after subtraction of the empty rotor signal should be shown (the rest in S1), so the reader can see how the fitted spectrum looks like and how it compares with experiment.

Response: We thank the reviewer for the comment. The fit of the rotor signal was reconsidered for all spectra in the manuscript. The following points have to be clarified:

- The rotor signal is relatively intense compared to the samples' signals, this is due to the small quantity of silanols in the samples with respect to the protons present in the Vespel caps (polyimide polymer).
- We have verified the source of this signal by recording the spectrum of an empty (clean and dry) rotor at different speeds. The observed peaks get narrower at higher MAS frequencies proving their provenance from the rotor and not from the stator (probe).
- The rotor signal is not exactly the same for all samples exhibiting different magnetic susceptibilities. This led us to consider small shift corrections, keeping the total intensity of the rotor bands the same in all fits (see Figure 1). This information is added in the captions of the figures, highlighted in yellow.

Figure 1. ^1H MAS NMR spectra of Cs-CHA sample (without CO_2) and the peaks related to the 1.9 mm rotor before and after rotor signal treatment. Color code: Black lines are the ^1H MAS NMR spectra of the rotor packed with Cs-CHA sample, the red lines are the rotor signals before and after correction (simulated by their corresponding deconvolution), and the hollow black peaks represent the rotor signal before and after the correction.

- We avoided the use of Hahn echo since it is suppressing the peaks with short relaxation times that may also come from strongly H-bonded silanols; for additional information please check our previous paper¹. Thus, to preserve the quantitative aspect of the spectra, single pulse experiments were exclusively performed. The figure below shows how two signals relaxing at different rates give rise to peaks with different widths. The use of a Hahn echo will not suppress the two species equally.

Figure 2. Schematic for the signals relaxing at different rates which give rise to peaks with different widths.

- In our previous paper¹, we have shown that H bonded silanols are not easy to quantify due to strong dipolar couplings. Thus, a rotation at high MAS frequency is a must despite the presence of the rotor signals, to suppress the dipolar couplings effects as much as possible.
- Please note that packed rotors were dehydrated under vacuum at 350 °C overnight before closing the rotors under Argon in a glove box; the NMR rotor was not contaminated. The experiments were repeated two times and the results were the same. The same treatment was done for the empty rotors.

1 E. Dib, I. M. Costa, G. N. Vayssilov, H. A. Aleksandrov and S. Mintova, *J. Mater. Chem. A*, 2021, 9, 27347–27352.

Comment 2: The second point concerns the interpretation of the data in Fig. 4, and hence the conclusions of this work. When reading Figure 4 it is clear that CO_2 uptake capacity depends primarily on the K^+/Cs^+ ratio. The smaller K^+ ion interacts stronger with CO_2 molecules allowing stronger electrostatic interactions at closer distances, and

being smaller and situated closer to the framework is in general less restrictive for diffusion through the material. So this is expected and generally known. On the other hand, the role of the silanol groups is less clear and this is the most interesting part of this work. Although there seem to be a linear trend, the final assessment on this has to wait until the robust ^1H MAS NMR data are available (see point 1).

Response: Please see the answer to the first comment regarding the ^1H MAS NMR data. In this work, we observed how a simple ion-exchange alters the silanols concentration. After CO_2 adsorption, the ^1H NMR spectra undergo slight shifts and broadenings, this led us to consider their interaction with CO_2 molecules. However, the cations are the main adsorption sites as shown in Figure 4. An increased overall H-bonding strength was also observed within all samples, see the ^1H MAS NMR while CO_2 was loaded (Fig S5). We further stressed only on the above observations in the modified manuscript.

Comment 3: In my opinion, DFT calculations included in this work are of little importance, since it is difficult to validate or relate these results to experimental data. On the other hand, it would be much more beneficial for this work to include ^1H NMR shifts calculations for different models to provide (in conjunction with experimental ^1H MAS NMR spectra) direct atomic-level insights into potential silanol groups environments in the material. I strongly encourage authors to do this; it would greatly elevate the impact of the paper when the experimental spectra could be directly related to distinct defects/structural motifs.

Response: The considered models were included to explain the difference in silanols concentration after ion exchange of the zeolite samples. The formation energy of the silanols is not the same in presence of different cations especially when the cage considered is an 8MR (Fig. 5 c). The DFT part has been revised. Firstly, CO_2 adsorption energies for different cases were calculated (Table S1). The results are in a good accordance with our previous work reporting on isosteric heat of CO_2 adsorption thus validating the models.¹ Proton chemical shifts with respect to TMS as a reference were also calculated (Table S2). All results suggested that after CO_2 adsorption, the proton chemical shift is disturbed. These findings were highlighted in the revised manuscript.

1 S. Ghosvandi, B. Coasne, E. B. Clatworthy, R. Guillet-Nicolas, P. Bazin, M. Desmurs, L. Jacobo Aguilera, V. Ruaux and S. Mintova, ACS Appl. Nano Mater., 2022, 5, 5578–5588.

Comment 4: I suggest some general improvements in the manuscript to improve clarity:

abstract, lines 19-21, “content” should rather be changed to ratios of different cations

page 4, lines 112-114, the sentence is long and difficult to follow, I think it should be improved, it is important that information about samples considered in the manuscript is presented clearly

Table 1, it is somehow awkward that molar absorption coefficients (that relate to IR) are presented after the NMR data, whereas IR data are presented first; NMR should be presented first instead, IR data are not quantitative anyway.

page 5, lines 127-131, “number of Si surrounding a SiO_4 site”, it is not clear for a non-NMR reader, should be explained clearly, as well as a shift dependence in case of Al neighbours.

Response: We thank you for all the constructive comments to improve the clarity of the main text. However, regarding your suggestions indicating firstly present the NMR data rather than FTIR. We believe that it is more convenient to present the FTIR results firstly since they cannot be used when CO_2 is adsorbed in the system. The limitation of FTIR in this study comes from the hard deconvolution of the spectra in the silanol region and the overlapping of the stretching modes of CO_2 in the same frequency range, once CO_2 was adsorbed. ^1H MAS NMR was used to further explore the silanol groups before and after CO_2 adsorption. Nevertheless, we applied all of your suggestions in the modified manuscript.

Reviewer #2:

“In this work, the authors investigated the influence of K^+ and Cs^+ on the distribution of silanol sites and the variations after the CO_2 adsorption by FTIR and NMR. They also revealed the relationship between total silanol

concentration and CO₂ adsorption capacities. The authors have done a systematic work concerning silanol sites properties and CO₂ adsorption with interesting findings. While some points should be addressed before its publication.”

Comment 1: Since silanol sites play a vital role in the CO₂ adsorption in this work, silanol sites are suggested to be added in the keyword.

Response: Thank you for the suggestion. *Silanol sites* was added as a keyword.

Comment 2: The first two paragraphs in introduction part are similar to the author's previous published work (ACS Appl. Nano Mater. 2022, 5, 4, 5578–5588).

Response: The two paragraphs were modified.

Comment 3: In the DFT part, the authors calculated the silanol formation energy of three cases, (i) silanol in a 6MR, (ii) silanol in confined corner of an 8MR next to a 4MR, and (iii) silanol in an 8MR. Before drawing the conclusion “In contrast, at the most accessible site in the 8MR, the formation of silanol defects is demonstrated to be more feasible in the presence of K⁺ than Cs⁺ cations” (page 11), the authors should give some evidences or cite literatures to prove the 8MR is the most accessible site for CO₂ rather the other two cases.

Response: Thank you for the comment. We modified the manuscript by citing our earlier work on localizing the position of CO₂ molecules inside the CHA structure using electron diffraction. In this work, we reported that CO₂ is located predominantly in the center of the 8MR at site SIII'. This position was also confirmed by Rietveld refinement of PXRD data.¹

1 M. Debost, P. B. Klar, N. Barrier, E. B. Clatworthy, J. Grand, F. Laine, P. Brázda, L. Palatinus, N. Nesterenko, P. Boullay and S. Mintova, *Angew. Chem. Int. Ed.*, 2020, 59, 23491–23495.

Comment 4: How does the author obtain the conclusion “The current theoretical/experimental findings show that Cs⁺ cations can also stabilize the CHA framework by ion-exchange insertion (Table 1, Fig. 4, and Fig. 5)” (page 11)?

Response: We followed the crystallization pathway of these nanosized CHA samples in our previous work by various different techniques such as microscopy, spectroscopy, elemental analysis, etc.¹ By following the changes in the concentration of the cations during the synthesis, we concluded that Cs⁺ cations are necessary to stabilize the CHA cages and to obtain long-ranged crystallinity in the samples.¹ This was also observed in this work by ion-exchange post-synthetic treatment. We modified the manuscript to emphasize on this result.

1 S. Ghojavand, E. B. Clatworthy, A. Vicente, E. Dib, V. Ruaux, M. Debost, J. El Fallah and S. Mintova, *J. Colloid Interface Sci.*, 2021, 604, 350–357.

Comment 5: It is indicated that “In small-pore zeolites with high cation contents, cage access is typically controlled by the size of the cation and the occupancy degree of the window.” (Nature Chemistry, 2013, 5, 89-90). Considering the large radius of Cs⁺, does the author think the occupancy of the cage window by Cs⁺ could be a reason for the decreasing trend of CO₂ adsorption capacity vs Cs⁺ amount?

Response: Increasing the size of the cations corresponds to reduction of available free voids within the zeolite framework which consequently decreases the CO₂ capacity. We showed this effect in our previous work.¹ However, in this work, we also observed that there is a change in CO₂ capacity when we have more silanol sites present in the structure through H-bonding; this also was observed by ¹H MAS NMR spectroscopy and approved by DFT calculations.

1 S. Ghojavand, B. Coasne, E. B. Clatworthy, R. Guillet-Nicolas, P. Bazin, M. Desmurs, L. Jacobo Aguilera, V. Ruaux and S. Mintova, ACS Appl. Nano Mater., 2022, 5, 5578–5588.

With best wishes,
Dr. Svetlana Mintova,
Directeur de Recherche, CNRS
LCS, ENSICAEN, Normandy University

Reviewers' comments:

Reviewer #1 (Remarks to the Author):

Dear Authors and Editor,

I appreciate all the changes and improvements implemented in the manuscript. It is better now and reads well. However, my skepticism regarding the reliability of the OH groups quantification prevails. The authors re-analyzed the NMR data by re-fitting the spectra, but my main point from the original report is still not addressed. Fitting these very minor signals in the presence of such severe background (with no flat baseline in the spectra for reference) is not reliable. As I understand, the authors tried subtraction of the empty rotor spectra, but it did not work, the subtraction was not perfect. Information about the fact that background cancellation in this “classical” way did not work should be included in the manuscript. The authors didn't record spin-echo acquisitions with claim that it may suppress those H species that exhibit fast relaxation. However, at the MAS rate of 40 kHz used by the authors, the rotor-synchronized spin-echo pulse sequence of two rotor periods would have the duration of only 50 μ s. It seems unreasonable to assume that in these diamagnetic systems signal components with that fast relaxation would occur. The biggest advantage of the spin-echo acquisition would be the improved baseline that would enable more reliable fitting. The authors focus attention of the reader on the very narrow spectral range from 3 to 12 ppm. The background extends much more than that, usually between -40 to 40 ppm. So with these figures in the manuscript it is difficult to assess how bad/severe the background really is, and where the baseline in the spectra is flat. Spectra in the wider range including sidebands should be presented in the SI. Note that even at the state-of-the-art acquisition conditions of 60 kHz MAS with double-adiabatic spin-echo sequence (Rzepka, Bacsik, Pell, Hedin, and Jaworski *J. Phys. Chem. C* 2019, 123, 35, 21497–21503) the background extends over several tens of ppm. So if the authors do not want to try spin-echo, and if that fails, to use a real background suppression technique (*J. Magn. Reson.* 332 (2021) 107067; *Solid State Nucl. Magn. Reson.* 57-58 (2014) 22-28; *Solid State Nucl. Magn. Reson.* 63-64 (2014) 13-19; *J. Magn. Reson.* 53 (1983) 365-385; *J. Magn. Reson.* 80 (1988) 128-132; *J. Magn. Reson.* 100 (1992) 336-341; *Solid State Nucl. Magn. Reson.* 26 (2004) 11-15; *J. Magn. Reson.* 209 (2011) 300-305; *J. Magn. Reson.* 221 (2012) 41-50) the claim about quantification of the OH groups has to be significantly downplayed in my opinion. The errors are expected to be very significant and this should be clearly indicated in the manuscript.

Dear reviewer,

The manuscript was revised considering the comments and the changes are highlighted in both the revised manuscript and supporting information attached.

Here is a point-by-point response to the reviewers' comments and concerns.

Reviewer #1:

"I appreciate all the changes and improvements implemented in the manuscript. It is better now and reads well. However, my skepticism regarding the reliability of the OH groups quantification prevails. The authors re-analyzed the NMR data by re-fitting the spectra, but my main point from the original report is still is not addressed. Fitting these very minor signals in the presence of such severe background (with no flat baseline in the spectra for reference) is not reliable. As I understand, the authors tried subtraction of the empty rotor spectra, but it did not work, the subtraction was not perfect. Information about the fact that background cancellation in this "classical" way did not work should be included in the manuscript. The authors didn't record spin-echo acquisitions with claim that it may suppress those H species that exhibit fast relaxation. However, at the MAS rate of 40 kHz used by the authors, the rotor-synchronized spin-echo pulse sequence of two rotor periods would have the duration of only 50 μ s. It seems unreasonable to assume that in these diamagnetic systems signal components with that fast relaxation would occur. The biggest advantage of the spin-echo acquisition would be the improved baseline that would enable more reliable fitting. The authors focus attention of the reader on the very narrow spectral range from 3 to 12 ppm. The background extends much more than that, usually between -40 to 40 ppm. So with these figures in the manuscript it is difficult to assess how bad/severe the background really is, and where the baseline in the spectra is flat. Spectra in the wider range including sidebands should be presented in the SI. Note that even at the state-of-the-art acquisition conditions of 60 kHz MAS with double-adiabatic spin-echo sequence (Rzepka, Bacsik, Pell, Hedin, and Jaworski J. Phys. Chem. C 2019, 123, 35, 21497–21503) the background extends over several tens of ppm. So if the authors do not want to try spin-echo, and if that fails, to use a real background suppression technique (J. Magn. Reson. 332 (2021) 107067; Solid State Nucl. Magn. Reson. 57-58 (2014) 22-28; Solid State Nucl. Magn. Reson. 63-64 (2014) 13-19; J. Magn. Reson. 53 (1983) 365-385; J. Magn. Reson. 80 (1988) 128-132; J. Magn. Reson. 100 (1992) 336-341; Solid State Nucl. Magn. Reson. 26 (2004) 11-15; J. Magn. Reson. 209 (2011) 300-305; J. Magn. Reson. 221 (2012) 41-50) the claim about quantification of the OH groups has to be significantly downplayed in my opinion. The errors are expected to be very significant and this should be clearly indicated in the manuscript."

Response:

We thank the reviewer for considering positively all the improvements implemented in the manuscript and for the variety of literature provided for background suppression techniques.

Regarding the quantification of silanols, we agree with the reviewer that the method used for the deconvolution of the spectra lacks perfection because of a considerable signal coming from the Vespel caps of the 1.9 mm rotors used in this experiment.

In addition, we are aware that the strong dipolar couplings for high chemical shift protons (3-8 ppm) already induce slight errors in the quantification of the species, as we have mentioned in the revised manuscript. This was revealed by correlating the chemical shifts and the H-bond length for silanols; please see the paper recently published in the Journal of Materials Chemistry A (Journal of Materials Chemistry A 9 (48), 27347-27352). The biggest changes occur for non-H bonded silanols (0-3 ppm). Despite all the limitations, we believe we can rely on the changes in the areas of the ^1H NMR spectra to quantify silanols in this study, and these results are in line with the areas of silanols regions in the IR spectra.

While this rotor signal is complicating the fitting procedure, we selected to keep using the direct acquisition using single pulse experiments to avoid losing wide peaks that may belong to the sample when using an echo experiment. However, we performed several Hahn echo experiments to answer reviewer's concerns and the results are presented in Fig. R1. In these experiments, the stator signal (outside of the coil) is entirely suppressed after a short '2 rotor periods' delay (50 μ s). On the other hand, the rotor signal, being in rotation (inside the coil), has a considerable relaxation time and it is not removed even after a delay of '20 rotor periods' (500 μ s): see Fig. R1. The presence of the rotor signal, even after '20 rotor periods', led us to consider only single acquisition

experiments, as it was used lately in several papers recently published (Journal of Materials Chemistry A 2021, 9 (48), 27347-27352, Inorganic Chemistry Frontiers 2022 9 (6), 1125-1133). Even if the spin-echo acquisition would allow improving the baseline that would enable more reliable fitting, the persistence of the rotor signal and its unknown relaxation behavior in different samples, lead us to consider the single acquisition exclusively here.

We have also acquired the spectra using anti ringing experiment (three successive 90° pulses with different phases) but still the peaks coming from the rotor caps persist, see Fig R1.

Fig R1. ¹H NMR spectra of AP-CHA sample acquired at MAS frequency of 40 kHz using different approaches.

Additionally, to check the behavior of the empty rotor under echo condition, a single acquisition of an empty rotor is compared to a Hahn echo acquired after a delay of '4 rotor periods'. These spectra are in accordance with our previous discussion (Fig. R1) and showed that Hahn echo experiment did not suppress the rotor signal but also concealed some parts of sample signals.

Fig R2. ^1H NMR spectra of empty rotor acquired at MAS frequency of 40 kHz using different approaches.

The following changes were made in the manuscript:

1. We included a footnote to stress that the classical subtraction of the rotor signal is not possible due to changes in magnetic susceptibility of the rotor (empty vs. filled): it is highlighted in the main text.

The changes in magnetic susceptibility of the rotor (empty vs. filled) induce slight chemical shifts variation in the spectra. The subtraction of the rotor signal is not a trivial task using the routine procedures available in Topspin software. Additional Hahn echo or anti-ringing experiments did not suppress the rotor signal but also concealed some parts of sample signals (see Figs. S7 and S8). So, for better resolution, only single pulse experiments are discussed in this work and the rotor signal was considered during the fitting procedure after small adjustments of chemical shifts while keeping the overall intensity of the rotor caps constant.

We included the new figures in the Supporting information as Figs. S7 and S8.

With best wishes,
 Dr. Svetlana Mintova,
 Directeur de Recherche, CNRS
 LCS, ENSICAEN, Normandy University

Reviewers' comments:

Reviewer #1 (Remarks to the Author):

Dear Authors, dear Editor,

It is great that the authors finally tested the suggested spin-echo acquisition. Thank you. As expected, the improvements when using spin-echo are dramatic. When comparing single pulse experiment (blue trace) in Figure R1 with the short spin-echo (2 rotor periods; red trace) it is clear that in the short spin-echo experiment already 90% of the broad background signal is suppressed. In case of the longer spin-echo (20 rotor periods; green trace) the spectrum is essentially free from any broad background signals and the baseline is flat, which makes possible to fit the spectrum. The true signals from the sample seem to be unaffected by the spin-echo, despite the apparently lower intensity, which actually originates from the suppression of the broad background components that were present beneath. In my opinion, these results render spin-echo experiments as the only reliable way of assessing weak proton signals from these samples. As I indicated in the first-round report, fitting these small signal components in the presence of such severe background as it is presented now in the manuscript is unreliable, because the baseline is not defined in the fitting region. Yet, it is even worse than I thought initially, since the shifts of the background signals change, and it has to be accounted for during fitting. It deteriorates the accuracy even further.

Therefore, as suggested in the original report, I suggest to collect and fit the spin-echo spectra. In my opinion it would be great to present these new (reliable) data together with the current "brut force" fitting results of the single pulse spectra. In this way the manuscript will not only be strengthened by inclusion of the reliable estimates of the silanol groups concentrations, but will also gain a methodological/educational role in the field, by indicating the spread of the values depending on the experimental/fitting method chosen.

Dear reviewer,

The manuscript was revised considering the comments and the changes are highlighted in the revised manuscript and supporting information attached.

Here is a point-by-point response to the reviewers' comments and concerns.

Reviewer #1:

"It is great that the authors finally tested the suggested spin-echo acquisition. Thank you. As expected, the improvements when using spin-echo are dramatic. When comparing single pulse experiment (blue trace) in Figure R1 with the short spin-echo (2 rotor periods; red trace) it is clear that in the short spin-echo experiment already 90% of the broad background signal is suppressed. In case of the longer spin-echo (20 rotor periods; green trace) the spectrum is essentially free from any broad background signals and the baseline is flat, which makes possible to fit the spectrum. The true signals from the sample seem to be unaffected by the spin-echo, despite the apparently lower intensity, which actually originates from the suppression of the broad background components that were present beneath. In my opinion, these results render spin-echo experiments as the only reliable way of assessing weak proton signals from these samples. As I indicated in the first-round report, fitting these small signal components in the presence of such severe background as it is presented now in the manuscript is unreliable, because the baseline is not defined in the fitting region. Yet, it is even worse than I thought initially, since the shifts of the background signals change, and it has to be accounted for during fitting. It deteriorates the accuracy even further.

Therefore, as suggested in the original report, I suggest to collect and fit the spin-echo spectra. In my opinion it would be great to present these new (reliable) data together with the current "brut force" fitting results of the single pulse spectra. In this way the manuscript will not only be strengthened by inclusion of the reliable estimates of the silanol groups concentrations, but will also gain a methodological/educational role in the field, by indicating the spread of the values depending on the experimental/fitting method chosen."

Response:

We conducted a study to investigate the effect of post-synthetic ion-exchange on nanosized chabazite (CHA) zeolite, focusing on the formation of silanol sites. We found that during the ion-exchange treatment, silanol sites of CHA altered significantly. We also observed a redistribution of silanol sites strength (i.e. the hydrogen bonding strength) upon adsorption of CO₂ on these samples. The particular study of silanol sites was possible using FTIR and high resolution ¹H MAS NMR spectroscopies. These results are in good agreement with each other and they correlated with DFT calculations. The experimental and theoretical results we obtained were consistent with the conclusions we drew in the paper.

We are highly interested to address the reviewer's suggestion to use spin echo experiments for quantification of silanols, and we acknowledge the potential benefits of this technique. However, we must note that this is not the focus of this paper. We kindly highlight the main focus of this manuscript: **to investigate the effect of post-synthetic ion-exchange on nanosized chabazite (CHA) zeolite focusing on the formation of silanol sites**. This is addressed in details in the current form of our paper via different methods. Additionally, the current fitting approach we employed is discussed and proven to be reliable in many respected publications (see selected papers 1-4). To further support our fitting approach, we already presented new results in the previous response to the reviewer's document and the supporting information of the paper (re-presented here in Fig. R1 and Fig. R2). **Fig R1 and Fig. R2 clearly show that employing spin echo experiments not only is unable to remove all the rotor signal, but also hinders a part of the sample signals**. Thus, based on these reasons, we highly believe that our fitting approach is more suitable in this study.

Fig R1. ^1H NMR spectra of AP-CHA sample acquired at MAS frequency of 40 kHz using different approaches.

Fig R2. ^1H NMR spectra of empty rotor acquired at MAS frequency of 40 kHz using different approaches.

We would like to highlight that direct comparison between spin echo method vs. our fitting approach does not lie within the scope of the present manuscript. **We believe that this direct comparison could be of great interest for educational purposes. We are highly willing to discuss the possibility of writing an educational paper on this topic with the reviewer.** We agree that the use of different experimental and theoretical methods is

important for advancing the field, and we would welcome the opportunity to explore this further in a separate paper.

If the reviewer is interested in pursuing this idea, we would be happy to collaborate on such a paper and further discuss the details with them.

However, we have edited the subsection entitled "Characterization" detailing the technical limitations we are facing to quantify silanols and discussing the eventual sources of error. This was also added in the caption of Fig 3.

The highlighted revised manuscript and Supporting Information are attached.

Selected references:

1. Dib E, Costa IM, Vayssilov GN, Aleksandrov HA, Mintova S. Complex H-bonded silanol network in zeolites revealed by IR and NMR spectroscopy combined with DFT calculations. *J Mater Chem A*. 2021 Dec 14;9(48):27347–52.
2. Dubray F, Dib E, Medeiros-Costa I, Aquino C, Minoux D, Daele S van, et al. The challenge of silanol species characterization in zeolites. *Inorg Chem Front*. 2022 Mar 15;9(6):1125–33.
3. Ghojavand S, Dib E, Riodent B, Magisson A, Ruaux V, Mintova S. Altering the Atomic Order in Nanosized CHA Zeolites by Postsynthetic Silylation Treatment. *Adv Sustain Syst*. n/a(n/a):2200480.
4. Vayssilov GN, Aleksandrov HA, Dib E, Costa IM, Nesterenko N, Mintova S. Superacidity and spectral signatures of hydroxyl groups in zeolites. *Microporous Mesoporous Mater*. 2022 Sep 1;343:112144.

With best wishes,
Dr. Svetlana Mintova,
Director of Research, CNRS
LCS, ENSICAEN, Normandy University